# Effect of different lithological assemblages on shale reservoir properties in the Permian Longtan Formation, southeastern Sichuan Basin: Case study of Well X1

**Qian Cao**[1,2]*, **Xin Ye**[1,3], **Yan Liu**[4], **Pengwei Wang**[1,3], **Ke Jiang**[4]

**1** State Key Laboratory of Shale Oil and Gas Enrichment Mechanisms and Effective Development, Beijing, China, **2** Shale Gas Evaluation and Exploitation Key Laboratory of Sichuan Province, Chengdu, China, **3** Research Institute of Petroleum Exploration & Production, SINOPEC, Beijing, China, **4** College of Energy and Resources Chengdu University of Technology, Chengdu, Sichuan Province, China

* 421664225@qq.com

**Data Availability Statement:** All relevant data are within the paper and its Supporting information files.

## Abstract

Various types of marine-continental transitional facies are present in the gas-bearing shales of the southeastern Sichuan Basin. A review of the different lithological assemblages in these rocks is important for assessing the likely shale gas content and the development of the storage space. This study of the lithological assemblages of the Permian Longtan Formation in the southeastern Sichuan Basin at Well X1 used core observations, optical thin-section observations, Ar-ion polishing, scanning electron microscopy, and nitrogen adsorption tests to compare and analyze storage space types and pore structures in the shale to determine the sedimentary paleoenvironment, petromineralogy, and organic content. The marine-continental transitional facies in the study area were deposited in a warm climate that favored enrichment by organic matter. The kerogen is type $II_2$-III (average vitrinite reflectance 2.66%), which is within the favorable thermal maturity range for the presence of shale gas. The lithology mainly consists of shale, siltstone, and limestone (with bioclasts), as well as a coal seam. The lithological development divides the Longtan Formation into lower (swamp), middle (tidal flat/lagoon), and upper (delta) sub-members. From lower to upper divisions, the lithofacies evolved from silty shale to clay shale and then to shale intercalated with siltstone or calcareous layers. The proportions of intergranular and dissolution pores in the clay minerals decrease gradually from lower to upper sub-members, and pore size sizes also tend to decrease. Relatively large-diameter pores and microfractures occur in the inorganic matter in the lowest sub-member. Quartz and clay are the main constituents of the shale, respectively contributing to the specific surface area and specific pore volume of the reservoir space.

## 1. Introduction

During multi-cycle tectonic evolution and deposition in the Permian in China, several sets of organic-rich shale systems developed with marine, marine-continental transitional, and continental facies [1]. Shale gas is abundant in the marine-continental transitional facies,

**Funding:** The author(s) received no specific funding for this work.

**Competing interests:** The authors have declared that no competing interests exist.

amounting to approximately $19.8 \times 10^{12} \, \text{m}^3$, or 25% of the total shale gas reserves in China, and represents extensive exploration and development potential [2]. The presence of mainly Carboniferous-Permian marine-continental transitional shale gas had previously been confirmed in the Ordos, Sichuan, Bohai Bay, and Qaidam Basins [3–5]. The Sichuan Basin, located at the western margin of the upper Yangtze plate, has one of the highest potentials for natural gas exploration and development in China. The Sichuan Basin underwent complex geological structural development from the Paleozoic to the Cenozoic, evolving from a craton basin to a foreland basin with complex superimposed basins containing both marine and continental facies [6, 7]. The source rocks in the Longtan Formation occur widely throughout the Sichuan Basin, mainly in the east, southeast, and southwest. Many wells in the Sichuan Basin and the surrounding late Permian Longtan Formation have encountered varying amounts of gas in the shale, indicating favorable gas prospects [8–12]. The sedimentary facies of the Longtan Formation display obvious transformations from continental to marine facies (i.e., fluvial / swamp / tidal flat / shallow-water shelf / deep-water shelf) located in an arc stretching from the southwest to the northeast [13, 14].

The Longtan Formation shale strata in the southeastern Sichuan Basin encompass both thin layers and large accumulated thicknesses containing a number of coal seams or coal lines, characteristically with frequently interbedded shale, coal, sandstone, and limestone [15]. Transitional facies reservoirs such as these differ from, for example, the Longmaxi Formation shale in terms of organic matter types and content. Marine-continental transition shale has attracted increasing attention in China in recent years, but relatively few studies have been reported on its reservoir-forming properties. Although the present comparative study of the pore structure of marine shale provides a certain understanding in key areas, comparative studies of the pore structure of shales in different sedimentary facies have yet to be carried out [16–20].

The lithological assemblages in the different thicknesses in the Longtan shale play a key role in hydrocarbon accumulation, storage, and migration [8, 21–24]. The presence of many different lithologies and frequent depositional superposition, as well as complex lithofacies transformations, present a difficult obstacle to determining the mechanisms and main influences on shale gas enrichment, and therefore on the prediction of favorable exploitation areas [15]. To date, research on shale gas in the Longtan Formation has mainly focused on gas generation potential and gas-bearing capacity [8, 25, 26] but few studies of reservoir development in different lithological assemblages are available; in particular, the exploration potential remains to be investigated [27]. Studies of shale reservoir development in the Longtan Formation, especially the relationship between the different lithological assemblages, should be strengthened to provide a theoretical basis for gas exploration.

Numerous investigative methods and technologies were used in the present study: X-ray diffractometry (XRD), organic geochemistry, organic maceral identification, Ar-ion polishing, scanning electron microscopy (SEM) and $N_2$ adsorption testing. In addition, comparisons and analyses of the different lithological assemblages and sedimentary properties of the Longtan Formation shale were conducted to define the properties of the reservoirs that developed over time in the different sedimentary environments described above. The findings lay a foundation for further exploration and development in the study area.

## 2. Materials and methods

### 2.1 Samples

After the deposition of the middle Permian, due to the influence of the Soochow movement, the seawater retreated eastward. This resulted in the rise of the western part of the Sichuan Basin, which became land, forming the "west land and East China Sea" paleogeographic

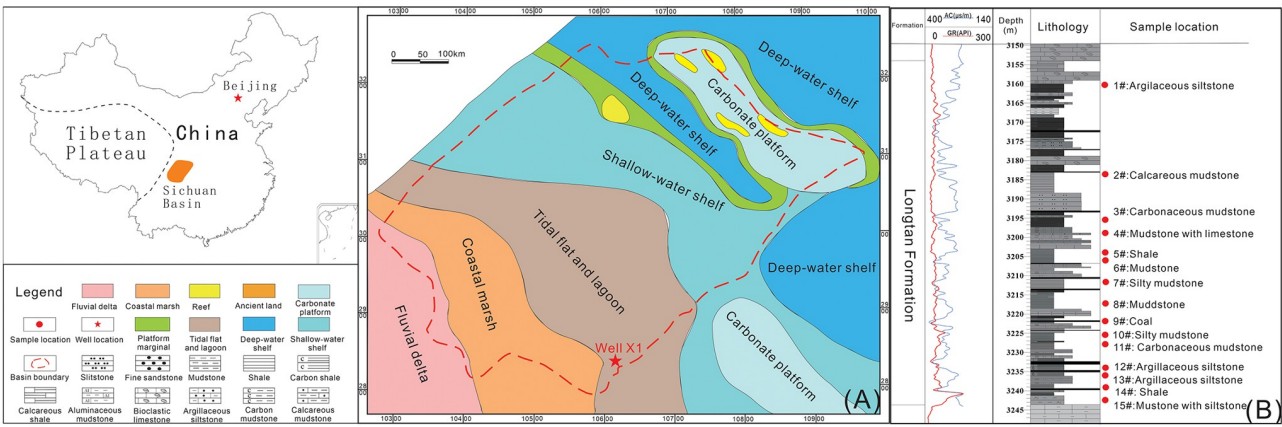

**Fig 1. Geographical location, sampling location, and stratigraphic description of the Longtan Formation in the study area: (A) Sample location; (B) Well X1 in Longtan Formation (modified from [30]).**

pattern which is high in the southwest and low in the northeast (Fig 1A) [6, 11]. It is clear that at the beginning of the late Permian, the distribution of sedimentary facies in the basin evolved from continental to marine facies from west to east [8, 13]. The Longtan Formation in the southern Sichuan Basin is a series of widely deposited coal-bearing rocks with a thickness of 20–120 m in parallel unconformable contact with the Maokou Formation. From land to sea, the sedimentary facies belt spreads roughly west-east, extends in a north-south direction, and mainly developed a tidal flat-lagoon sedimentary system [24, 28–30]. The water was relatively shallow, with a warm and humid tropical rainforest climate, and the development of peat swamps favored the growth, deposition, and preservation of terrestrial plants, making it ideal for the enrichment of organic matter.

Fifteen samples were taken from Well X1 at the southeastern margin of the Sichuan Basin, located in the low-steep fold belt, southern Sichuan Basin. The lithology of the Longtan Formation is mainly black shale and carbonaceous shale, with coal seams developing (Fig 1B) [15].

## 2.2. Methods

Total organic carbon (TOC) content was measured using a RJXWK-1 carbon-sulfur analyzer in accordance with the TOC test standard for sedimentary rocks (GB/T 19145–2003). The thermal evolution maturity (*Ro*) of the organic matter was measured using a Zeiss Axio Scope A1 polarized light microscope and a TIDAS S MSP 400 spectrophotometer (J&M Analytik AG) in accordance with the standard vitrinite reflectance test procedure for sedimentary rocks (SY/T 5124–2012). Organic matter types were determined using a Zeiss Axio Scope A1 polarized light microscope in accordance with the standard method of kerogen maceral identification and classification (SY/T 5125–2014). The macerals, morphology, and structure of kerogen were observed under transmitted light and fluorescence. Because the kerogen extraction process may destroy the structure of organic matter and the developmental state of some organic matter in shale and thereby affect the determination of the original source of the organic matter, the samples were observed in thin section [31].

A qualitative analysis of the mineral composition was conducted using a Zeiss Axio Scope A1 polarized light microscope in accordance with the SY/T 5368–2016 thin-section identification standard. A quantitative analysis of mineral composition was carried out on the X'Pert

Powder X-ray diffractometer (Malvern Panalytical) in accordance with the standard XRD analysis method for clay and common non-clay minerals in sedimentary rocks (SY/T 5163–2010).

SEM testing was performed on a Zeiss Sigma 300 scanning electron microscope in accordance with the standard SEM procedure for rock samples (SY/T5162-2014), and the images were analyzed using image analysis software (Image-pro Plus) to determine the quantitative characterization of pore structure parameters (e.g., the number, diameter, width, perimeter, and area) [32, 33].

Nitrogen adsorption testing was used to determine the specific surface and pore size distribution of the rock using the static nitrogen adsorption volume method (Micrometrics ASAP 2460 automatic specific surface and pore size analyzer) to complete the test process in accordance with GB/T 19587-2017/ISO 9277: 2010.

The content of major elements was determined using an AXios mAX X-ray fluorescence spectrometer (Malvern Panalytical) in accordance with the GB/T 14506.28–2010 standard. Testing accuracy was determined to be better than ± 3%.

## 3. Results

### 3.1. Lithological assemblages and sedimentary characteristics

In the study area, the carbonaceous shale contains some coal. From bottom to top, the evidence indicates that the Longtan Formation experienced a low energy in static water → medium-low energy → relatively low-energy sequence [13, 34]. The lateral distribution of the formation in the southeastern Sichuan Basin is basically stable, with a strata thickness between 70 and 80 m, but the lithological content varies greatly within the different thicknesses [29]. The shale is mainly gray-black and black, with thin coal seams (perhaps better described as 'coal lines') and carbonaceous shale mixed with limestone, marlstone, silty-fine sandstone, coal lines, and some siderite. Many such shale beds occur, all with numerous interbedded layers [15, 30, 35].

The mineral composition contents of 15 samples from Well X1 were measured by XRD whole-rock and clay mineral tests. The differences in the lithological assemblages showed that the Longtan Formation could be further divided into Lower, Middle, and Upper Longtan members (Table 1).

The Upper member is dominated by cross- and parallel-bedded deltaic sediments and contains plant fossils. Delta front subfacies are evident in this section. The lithology consists mainly of mudstone bioclasts, shale mixed with limestone or siltstone, and occasional coal lines (Table 1, Fig 2A–2D), and Bivalve and Bryophyte bioturbation, horizontal bedding and nodular siderite are observed in the core (Fig 2E). Clay minerals predominate (average content 44.5%), with about 33.1% carbonate minerals (calcite and dolomite) and about 18.0% quartz

**Table 1. Comparison of lithological combinations in the Permian Longtan Formation, Well X1, southeastern Sichuan Basin.**

| Formation | Upper Longtan | Middle Longtan | Lower Longtan |
|---|---|---|---|
| **Depositional model** | Delta | Tidal flats and lagoons, developing swamp facies at the top | Tidal flats and lagoons, developing swamp facies at the bottom |
| **Subfacies** | Delta | Lagoon facies mainly, with some tidal flat-swamp facies | Developing tidal flats, swamp facies, some lagoon facies |
| **Microfacies** | Peat swamp | Lagoon mud, intertidal mixed flats, peat swamps | Tidal flats, peat swamps, lagoon mud |
| **Lithological combination** | Limestone with bio-clastic, siltstone with mud, silty shale, occasional thin coal seam | Thick shale with mainly limestone and silty shale, some shale and sandstone interbedded, and developing coal seams and nodular siderite | Shale interbedded with mainly siltstone and limestone, developing thick carbonaceous shale and thick coal seams, and siderite distributed as grains and clumps |

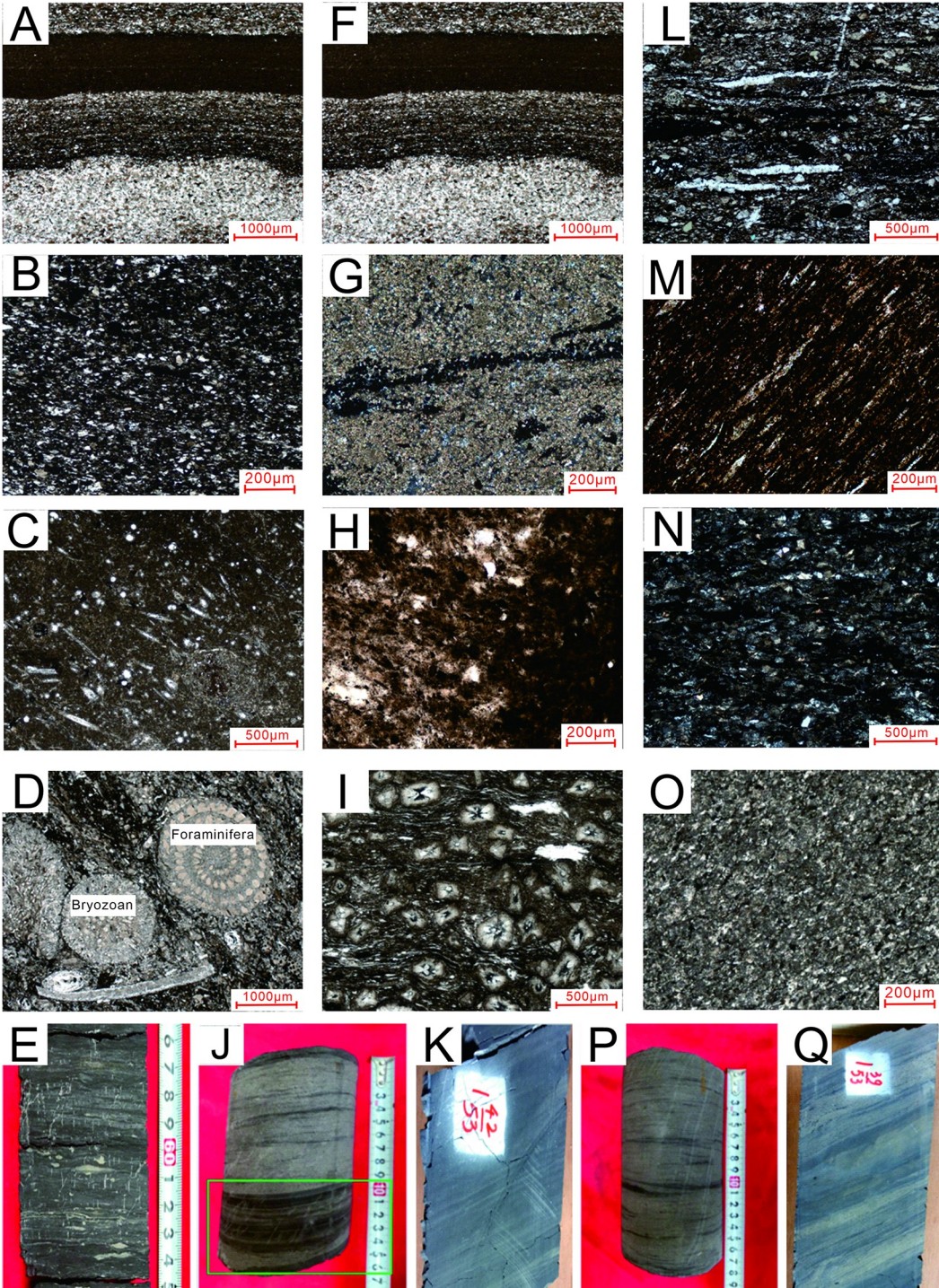

**Fig 2.** Thin section observations of lithological combinations in the Permian Longtan Formation, Well X1, southeastern Sichuan Basin: (A) Sample 3#: Argillaceous siltstone. The aggregation of clay minerals is mainly banded, and organic matter is clumpy and fine-banded, 5×10 (-); (B) Sample 7#: Shale. Organic matter is distributed along the bedding, 5×10 (-); (C) Sample 9#: Silty shale. Silty lamination develops, and organic matter is black-speckled, clumpy, and banded, 5×10 (-); (D) Sample 3#: Siltstone with mud. Organic matter is black, clumpy, and banded, 5×10 (-); (E) Bivalves and bryophytes. Bioturbation, horizontal bedding, nodular siderite; (F) Sample 4#: Shale with dolomite. Organic matter develops bedding, 10×10 (+); (G) Sample 6: Shale with siltstone. Bedding development, and organic matter develops bedding, 10×10 (-); (H) Sample 3#: Shale with bioclasts. Bioclasts are mainly spongy spicules scattered in rocks in needle-like and elliptical shapes, 5×10 (+); (I) Sample 8#: Carbonaceous mudstone. The mudstone is yellow-brown and is mainly an aggregation of

microcrystalline and cryptocrystalline clay minerals. A few cracks are filled with organic matter, 10×10 (-); (J) Horizontal bedding; (K) Wave-ripple bedding; (L) Sample 9#: Silty shale, 5×10 (+); (M) Sample 14#: Bioclastic limestone. Bioclasts such as foraminifers and bryozoans were distributed elliptically, most of them metasomatized by calcite and siliceous minerals. A small number of ostracoda fragments were found scattered in the shape of tiny crescents, 5×10 (+); (N) Sample 6#: Mudstone. Siderite is nodular, spherical, with an inner radial structure, and overall characteristics of orthogonal extinction; and in sizes from 0.1–1.0 mm, 10×10 (+); (O) Sample 9#: Argillaceous limestone. Siderite mainly consists of fine microcrystalline granular and clumpy aggregates, 5×10 (+); (P) Lenticular bedding; (Q) Veined bedding and cross-bedding.

and feldspar clastics (Fig 3). The sedimentary environments of the Middle and Lower Longtan members were dominated by tidal flats, lagoons, and some swamps, with wavy bedding, horizontal bedding, and minor cross-bedding. Banded and nodular siderite occur together. All these reflect low-energy sedimentary environments, relatively deep water, and a warm climate. The lithology of the Middle Longtan is mainly thick shale or carbonaceous mudstone, with a relatively thick coal seam (Table 1, Fig 2F–2I). The average clay mineral content is 52.6%; the average content of clastic minerals (quartz and feldspar) is about 23.2%. Horizontal bedding and wave-ripple bedding are observed in the core (Fig 2G and 2K). In the Lower Longtan, mudstone is mainly interbedded with siltstone and limestone, and thin coal seams are developed. The average carbonate mineral content is about 22.2% (Fig 3). Siderite is evident as nodules and powdery crystals. The lithology distribution is mainly represented by interbedded mudstone, siltstone and limestone, and some thin coal seams occur in the Lower Longtan member (Table 1, Fig 2I–2L). The average content of clay minerals is 51.13%; the average content of clastic minerals such as quartz and feldspar is 30.73%; and the average mineral content of carbonate rock is 13.4% (Fig 3). Carbonate rock minerals are mainly siderite, which is mostly spherical and nodular. Phenomena such as lenticular bedding, veined bedding, and

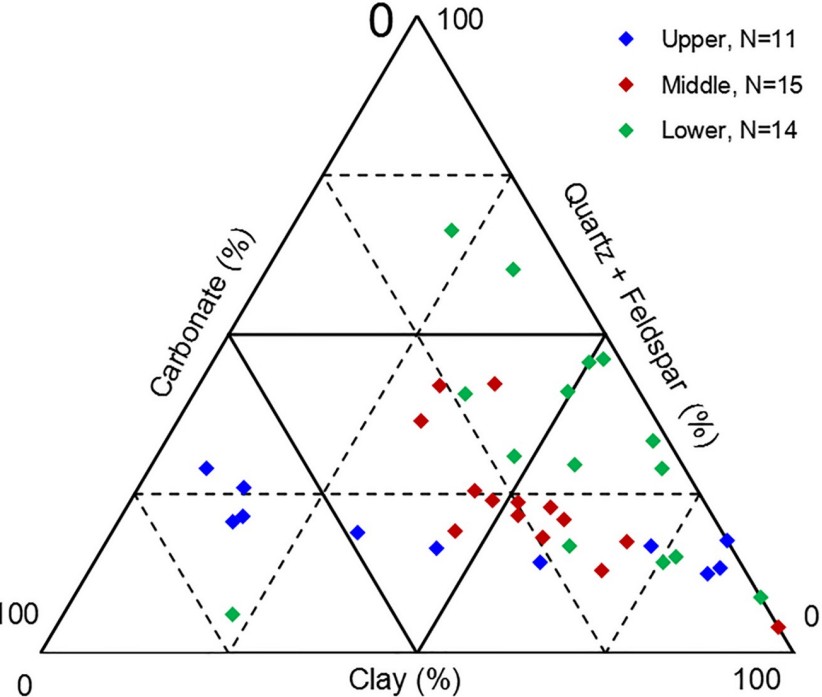

**Fig 3. Triangular diagram of mineral composition of Permian Longtan Formation samples from southeastern Sichuan (Note: 25 of these data points from [30]).**

cross-bedding, are obvious in the core (Fig 2P and 2Q). The observed aluminous mudstone from the Lower Longtan member is in unconformable contact with the limestone of the Maokou Formation.

## 3.2. Organic geochemistry characteristics

The type of organic matter not only determines the hydrocarbon generation capacity of source rocks and the properties of hydrocarbons, but also affects the occurrence and migration of gas in shale and has an important influence on the development of different types of reservoir space [8, 11, 15]. The analysis results (Table 2) for the organic matter types of eleven samples in Longtan Formation show that the kerogen types are mainly type III and type II$_2$-III and the main macerals are exinite and vitrinite, with mean contents of 56.09% and 36.45% respectively, indicating that the organic matter was mainly formed by fragments of reproductive organs and epidermal tissues of terrestrial higher plants after intense degradation, having clastic and/or partly flocculent shapes. Occasionally, some vitrinite is attached to surrounding amorphous humus. Organic matter types in different lithologies are different. Under a polarizing microscope, the maceral structure of different shale samples is given priority over the original structure, with a massive structure and small scattered plant debris. Vitrinite with a woody structure is commonly developed in coal samples and is mostly banded or clumpy, suspected to be formed by degradation of stems, leaves, lignocellulose, or some marine lower plants (Fig 4).

The thermal maturity differences are slight in the shale samples of the Longtan Formation according to the vitrinite reflectance test. The vitrinite reflectance ranges from 1.76% to 3.61%, with an average of 2.66%, and the samples are all in the high-maturity stage.

The distribution characteristics of organic matter are affected by paleoclimate, paleoproductivity, water redox conditions, and depositional rate. The proposed Chemical Index of Alteration (CIA) was used to compare and analyze the paleoclimatic conditions during the shale deposition period of the Permian Longtan Formation in Well X1 [36]; the calculation formula is as follows:

$$CIA = \left[ \frac{Al_2O_3}{CaO^* + Al_2O_3 + Na_2O + K_2O} \right] \times 100 \qquad (1)$$

where oxide units are moles, and CaO$^*$ refers only to CaO in silicate minerals. In this paper,

**Table 2. Analysis results for kerogen type of Longtan Formation samples.**

| Sample | Formation | Lithology | Sapropel (%) | Exinite (%) | Vitrinite (%) | Inertinite (%) | Type Index | Type |
|---|---|---|---|---|---|---|---|---|
| 2# | Longtan Formation | Calcareous mudstone | - | 66 | 28 | 6 | 6.00 | II$_2$ |
| 3# | Longtan Formation | Carbonaceous mudstone | - | 62 | 33 | 5 | 1.25 | II$_2$ |
| 4# | Longtan Formation | Mudstone with limestone | - | 42 | 53 | 5 | -23.75 | III |
| 6# | Longtan Formation | Mudstone | - | 54 | 41 | 5 | -8.75 | III |
| 7# | Longtan Formation | Silty mudstone | - | 54 | 41 | 5 | -8.75 | III |
| 8# | Longtan Formation | Mudstone | - | 23 | 71 | 6 | -47.75 | III |
| 10# | Longtan Formation | Silty mudstone | - | 64 | 32 | 4 | 4.00 | II$_2$ |
| 11# | Longtan Formation | Carbonaceous mudstone | - | 30 | 65 | 5 | -38.75 | III |
| 13# | Longtan Formation | Argillaceous siltstone | - | 66 | 30 | 4 | 6.50 | II$_2$ |
| 14# | Longtan Formation | Shale | - | 74 | 24 | 2 | 17.00 | II$_2$ |
| 15# | Longtan Formation | Mudstone with siltstone | - | 82 | 16 | 2 | 27.00 | II$_2$ |

Note: "-" indicates that the contents are negligible.

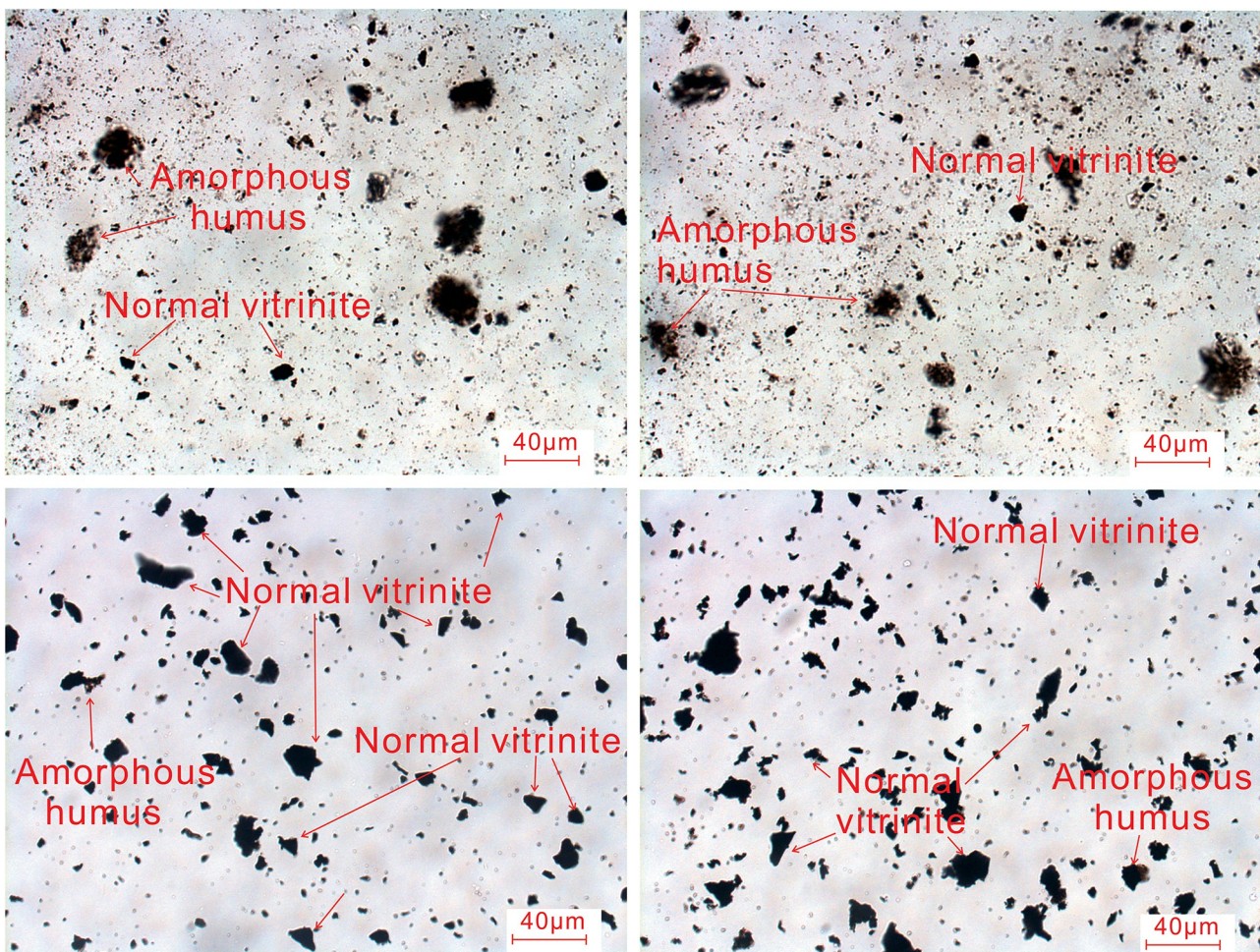

**Fig 4. Photomicrographs of typical organic material in Longtan Formation shale.**

CaO* is calculated indirectly from $P_2O_5$ using the following formula:

$$CaO^* = CaO - P_2O_5 \times 10/3. \tag{2}$$

In general, high CIA values indicate a warm and wet paleoclimate, whereas low CIA suggests a dry and cold paleoclimate. In particular, when CIA is between 50 and 65, it reflects a dry and cold climate under the background of low chemical weathering. CIA between 65 and 85 reflects a warm and wet climate under the background of moderate chemical weathering, and CIA between 85 to 100 reflects a hot and humid climate under the background of intense weathering [36].

The CIA of different samples from the Longtan Formation in the study area ranged from 77.85 to 84.86, with an average of 81.62 (Table 3), and there was no significant difference between the CIA values of the delta environment and the tidal flat-lagoon environment, indicating that the Longtan Formation shale was in a warm and wet climate during the deposition period. In addition, the rich deposits of reduction authigenic minerals (siderite) in the samples indicate a redox environment, which is conducive to the deposition of organic matter [37].

The TOC from the testing of the 15 samples from three wells in the Longtan Formation and the collection of 39 data points from references [13, 15, 30] indicate that the TOC distribution

**Table 3. Major element oxide contents of different samples from the Longtan Formation in the study area.**

| Minimum detectable value / Sample number | Major element oxide content (%) | | | | | | | | | | Ignition loss (%) | CIA(g/mol) |
|---|---|---|---|---|---|---|---|---|---|---|---|---|
| | SiO$_2$ | Al$_2$O$_3$ | MgO | Na$_2$O | K$_2$O | P$_2$O$_5$ | TiO$_2$ | CaO | TFe$_2$O$_3$ | MnO | | |
| | 0.0335 | 0.0375 | 0.0277 | 0.0021 | 0.0382 | 0.0019 | 0.0333 | 0.0438 | 0.0078 | 0.0138 | / | / |
| 1# | 40.51 | 13.47 | 2.26 | 0.52 | 1.84 | 0.17 | 2.60 | 11.52 | 8.02 | 0.10 | 18.34 | 84.01 |
| 2# | 33.57 | 14.16 | 2.89 | 0.74 | 0.91 | 0.31 | 2.83 | 7.73 | 15.89 | 0.15 | 18.01 | 79.60 |
| 3# | 42.62 | 24.25 | 0.83 | 1.07 | 1.34 | 0.33 | 4.57 | 0.70 | 6.60 | 0.02 | 17.37 | 84.86 |
| 4# | 38.04 | 23.34 | 1.02 | 1.10 | 0.90 | 0.21 | 5.08 | 0.75 | 12.43 | 0.17 | 16.62 | 83.78 |
| 5# | 43.35 | 24.07 | 1.33 | 1.10 | 0.96 | 0.36 | 4.57 | 1.58 | 7.53 | 0.06 | 14.22 | 80.87 |
| 6# | 40.30 | 23.19 | 1.15 | 1.45 | 1.40 | 0.40 | 4.79 | 0.86 | 9.58 | 0.11 | 16.32 | 81.94 |
| 7# | 41.71 | 15.07 | 3.43 | 0.63 | 0.48 | 0.37 | 4.20 | 5.46 | 16.22 | 0.09 | 9.81 | 81.20 |
| 8# | 35.27 | 10.01 | 3.83 | 0.15 | 0.20 | 0.43 | 1.70 | 11.80 | 19.84 | 0.16 | 15.00 | 81.72 |
| 9# | 40.64 | 19.15 | 2.12 | 1.38 | 1.32 | 0.42 | 3.11 | 4.12 | 11.90 | 0.16 | 14.80 | 83.56 |
| 10# | 36.34 | 12.73 | 2.61 | 0.59 | 0.52 | 0.18 | 1.59 | 1.08 | 28.45 | 0.30 | 15.81 | 77.85 |
| 11# | 40.92 | 19.39 | 1.74 | 1.14 | 1.63 | 0.41 | 3.66 | 2.48 | 12.28 | 0.16 | 15.44 | 79.90 |
| 12# | 38.92 | 23.07 | 1.09 | 0.87 | 1.97 | 0.06 | 4.10 | 0.62 | 14.45 | 0.04 | 14.47 | 82.84 |
| 13# | 39.58 | 22.98 | 0.66 | 1.17 | 1.46 | 0.21 | 3.78 | 0.69 | 10.15 | 0.20 | 19.53 | 79.69 |
| 14# | 41.36 | 25.39 | 0.47 | 0.62 | 0.83 | 0.23 | 4.55 | 0.50 | 6.06 | 0.11 | 20.11 | 79.00 |
| 15# | 37.94 | 19.62 | 1.35 | 0.59 | 3.02 | 0.45 | 3.36 | 3.51 | 14.37 | 0.15 | 13.11 | 81.10 |

of the Longtan Member is wide and bimodal, ranging from 0.15% to 83.02%, but mainly in the 2%–4% and 10%–20% ranges, with an average of 5.125% (Fig 5A). The TOC of the coal seams ranges from 20.49% to 83.020%. Because of the great variation in lithologies in the Longtan Formation, the TOC contents vary greatly in the vertical direction, showing strong heterogeneity. The TOC distribution of different samples was statistically analyzed taking the test results and lithological assemblage classification results into account. The TOC contents of 20 samples from the delta environment of the Upper Longtan member ranged from 0.15% to 20.49%, with an average of 3.5%. The TOC contents of eleven samples in the tidal flat-lagoon environment of the Middle Longtan member ranged from 0.17% to 66.62%, with a mean of 4.01%. The TOC contents of twenty-three samples in the tidal flat-lagoon environment of the Lower Longtan member ranged from 0.42% to 83.02%, with a mean of 9.37%. The variation in

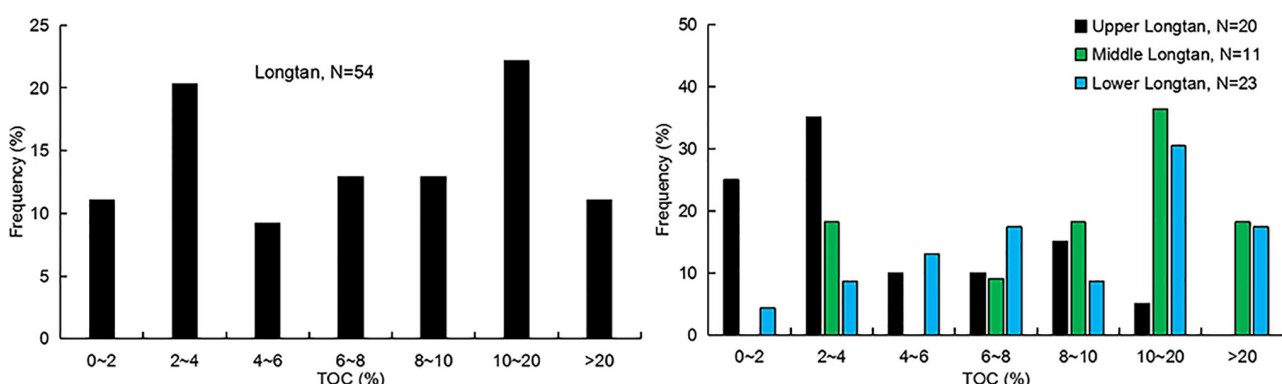

**Fig 5. Distribution of TOC contents of the Longtan Formation samples: (A) Distribution of all TOC contents; (B) Distribution of TOC in different sections of the Lower, Middle, and Upper Longtan members (data from this study and [13, 15, 30]).**

TOC content in lithological assemblages of different sedimentary environments is complex, with bimodal characteristics. The peak TOC values (TOC > 20%) of samples in the tidal flats and lagoons of the Middle and Lower Longtan members occur in coal and carbonaceous mudstone, whereas TOC contents are relatively low in argillaceous limestone and siltstone with mud of the Upper Longtan member. (Fig 5B).

### 3.3. Development characteristics of reservoir space

**3.3.1. Pore types.** Ar-ion polishing and field-emission SEM were used to observe the development characteristics of the microscopic storage space in samples from the Longtan Formation in the study area. Based on the pore-type classification of organic-rich shale [38] and the genetic and structural characteristics of shale pores, the main storage space of marine-continental transitional shale in the Longtan Formation was identified as consisting of inorganic pores, organic pores, and microfractures.

Inorganic pores are mostly distributed among clay mineral particles or in clastic particles such as quartz. Chemically unstable clay minerals (montmorillonite) produce numerous pores in the process of deposition and burial, and transform into andreattite or illite. Numerous interlayer pores and fractures develop in the lamellar andreattite aggregation, showing as

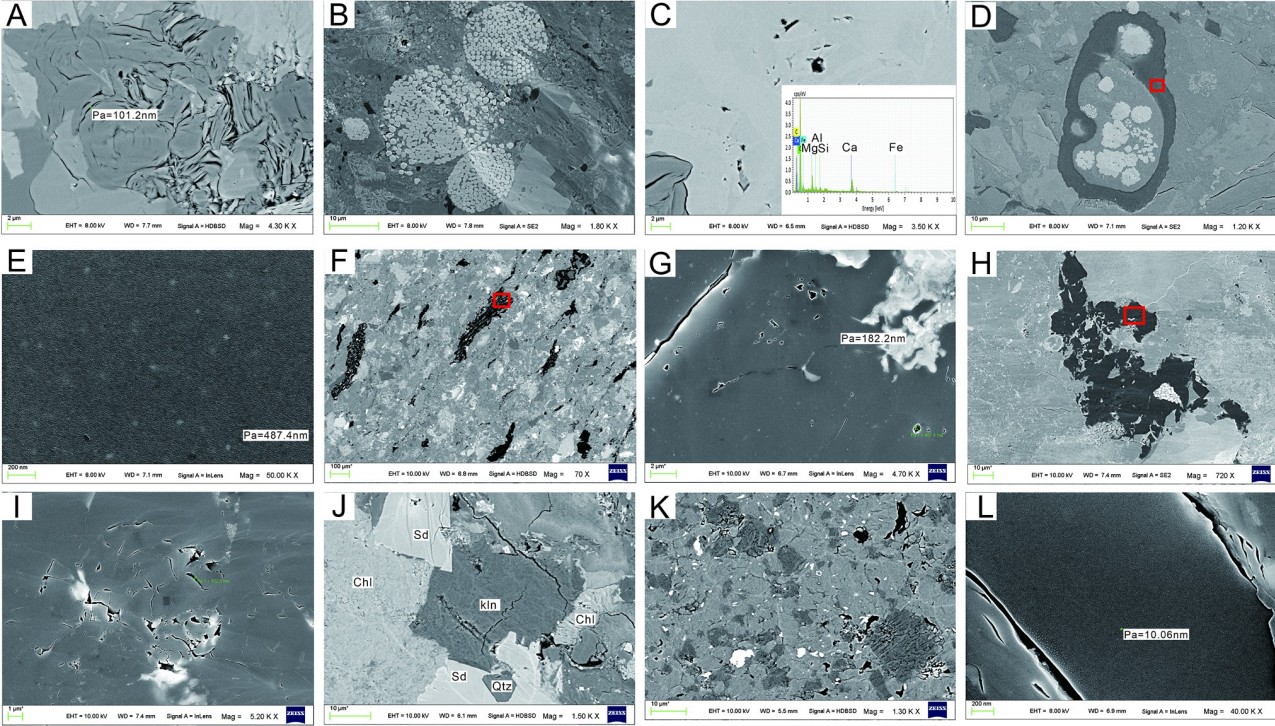

**Fig 6. Distribution characteristics of microscopic reservoir space in shale samples of the Longtan Formation.** (A) Sample 8#: Interlayer fractures developing in lamellar clay minerals (×4300); (B) Sample 2#: Framboidal pyrite aggregations showing parts of the intercrystal pore filled with clay minerals (×1800); (C) Sample 5#: Irregular dissolution pores in siderite (confirmed by energy spectrum) (×3500); (D) Sample 4#: Interior of crustal organic matter filled with micro-granular and framboidal pyrite, as well as clay minerals (×1200); (E) Enlarged view of Sample 4# in Fig 9D, showing the development of little pores (×50000); (F) Sample 9#: Interstitial organic matter interbedded with clay minerals and siderite (720×); (G) Enlarged view of Sample 9# in Fig 9F, with pores and fractures developed (×5200); (H) Sample 12#: The sample structure is tight, clastic, interbedded with clay minerals and organic matter, with fragmentary or interstitial organic matter (×70); (I) Enlarged view of Sample 12# in Fig 9H. Irregular pores and microfractures developed in fragmentary organic matter (×4700); (J) Sample 9: Microfractures (×1500), Sd: siderite, Chl: chlorite, Kln: kaolinite, Qtz: quartz; (K) Sample 12#: Micro-granular quartz in inlying contact interbedded with lamellar kaolinite aggregations; intergranular pores and fractures developed (×1300); (L) Sample 14: No pores observed in banded organic matter, but shrinkage fractures developed at edges of organic matter (×40000).

curved features (Fig 6A). The interbedded and intrastratal micropores not only create space for shale gas occurrence, but also provide microscopic migration channels for gas seepage. Some minerals (quartz and pyrite) are interfered with by the external environment during the self-growth process, resulting in the formation of intragranular pores in the crystal accumulation process (Fig 6B). The pore size varies from several nanometers to several hundred nanometers, with relatively low development degree and poor connectivity. Occasionally, intercrystal (intergranular) dissolution pores occur at the edge of siderite due to dissolution (Fig 6C), which are bay-shaped or long and narrow. Relatively few dissolution pores are observed under microscopic observation in the study area. Under a microscope, the occurrence patterns of organic matter in the Longtan Formation can be identified as filled, dispersed, and bedding enrichment. Some organic matter has certain structural characteristics, but with low-organic-matter pores, and some of these are isolated circular or oval (Fig 6D–6I). Microfractures are mainly developed in the intercrystal or interparticle spaces of clay minerals, but seldom at the edge of organic matter. They are mainly exogenous cracks and intergranular fractures formed under the action of external forces, generally with a width of 1–20 μm. Some microfractures cut straight through particles of clasolite or clay mineral (Fig 6J and 6K), and some shrinkage fractures are also observed between organic matter and mineral particles (Fig 6L), mainly due to volume shrinkage of organic matter during thermal evolution. The results show that the dominant storage spaces in different samples are pore fractures and micro-fractures between clay minerals; organic pores are poorly developed.

**3.3.2. Development characteristics of pore structures.** The porosities and pore structure development characteristics of different samples are measured by the helium gas method and low-temperature $N_2$ adsorption. The BET and BJH models are used to calculate the specific surface area and specific pore volume respectively. From the test results, the porosities range from 1.07% to 8.17%, but are mainly in the 2%–4% and 6%–8% ranges (Fig 7). Specific surface areas are mainly distributed from 1.122 $m^2$/g to 24.452 $m^2$/g, with an average of 11.619 $m^2$/g (Fig 7), and specific pore volumes range from 0.001 mL/g to 0.032 mL/g, with a mean of 0.016 mL/g (Fig 7).

The morphological characteristics of the hysteresis curve of the isothermal adsorption-desorption curve can qualitatively reflect the morphological characteristics of pores to some extent. However, the pore types in a shale reservoir are diverse, and the hysteresis curve is often the superposition of various typical curves. According to the classification of the IUPAC nitrogen adsorption and desorption isothermal curve, the morphology of the sample adsorption loop can be divided into three categories (Fig 8). The first adsorption loop (type I, represented by sample 13#) has the curve characteristics of both IUPAC types H2 and H3. Adsorption and desorption curves basically coincide in the low-pressure section ($0 \leq P/P_0 < 0.4$). An obvious inflection point and a wide hysteresis loop appear in the medium-pressure section ($0.4 \leq P/P_0 < 0.8$), indicating a complex pore structure and multiple pore types, and characterized by ink-bottle type pores. The slopes of the adsorption and desorption curves increase in the high-pressure section ($0.8 \leq P/P_0 < 1.0$), indicating the development of fluted open pores in macropores. The morphology of the second adsorption loop (type II, represented by sample 9#) is similar to a type I adsorption loop, but the difference is that the slopes of the adsorption and desorption curves do not increase significantly in the high-pressure section, indicating that the number of fluted open pores, mainly ink-bottle type pores, is less. The third adsorption loop (type III, represented by sample 3#) is similar to IUPAC type H4 pores. The adsorption/desorption curves in the low-pressure and medium-pressure sections almost overlap, and the inflection point is not obvious, indicating that the pores are single and mostly slit micropores.

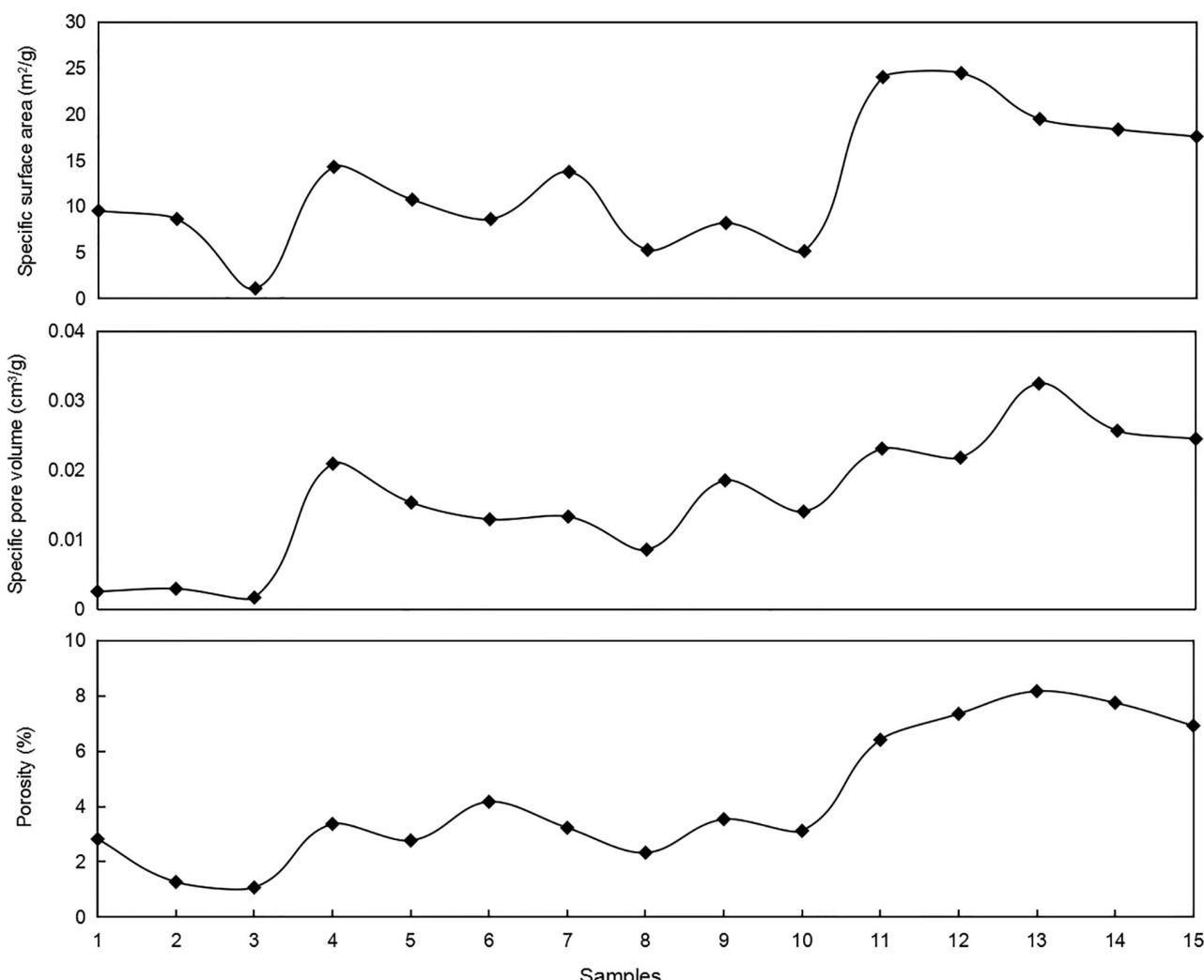

**Fig 7. Distributions of porosity, specific surface area, and specific pore volume of different samples.**

## 4. Discussion

The microscopic pore types, occurrence state, and enrichment degree of shale gas are controlled by sedimentary facies, TOC content, mineral composition, and tectonic evolution [1, 34, 39]. The sedimentary environment is one of the external factors controlling the formation of reservoir storage space, whereas the lithology and mineral composition, which are controlled by the sedimentary environment, are the internal factors controlling pore development. Therefore, the control effect of different lithological assemblages on shale pores is of great significance [40–42]. Image analysis software was used to identify the number, size, and face ratio of pores in different shale samples of the Longtan Formation [32]. The results, when combined with those of the $N_2$ adsorption and desorption test, indicate that the distribution of pore-structure characteristic parameters in different samples varies significantly. In the mudstone samples with silty and calcareous mudstone from the Upper Longtan member, the number of pores is relatively few, mainly inorganic pores, with pore size less than 20 nm (Fig 9A). The specific surface area and pore volume of the samples are relatively small (Fig 9B), which are

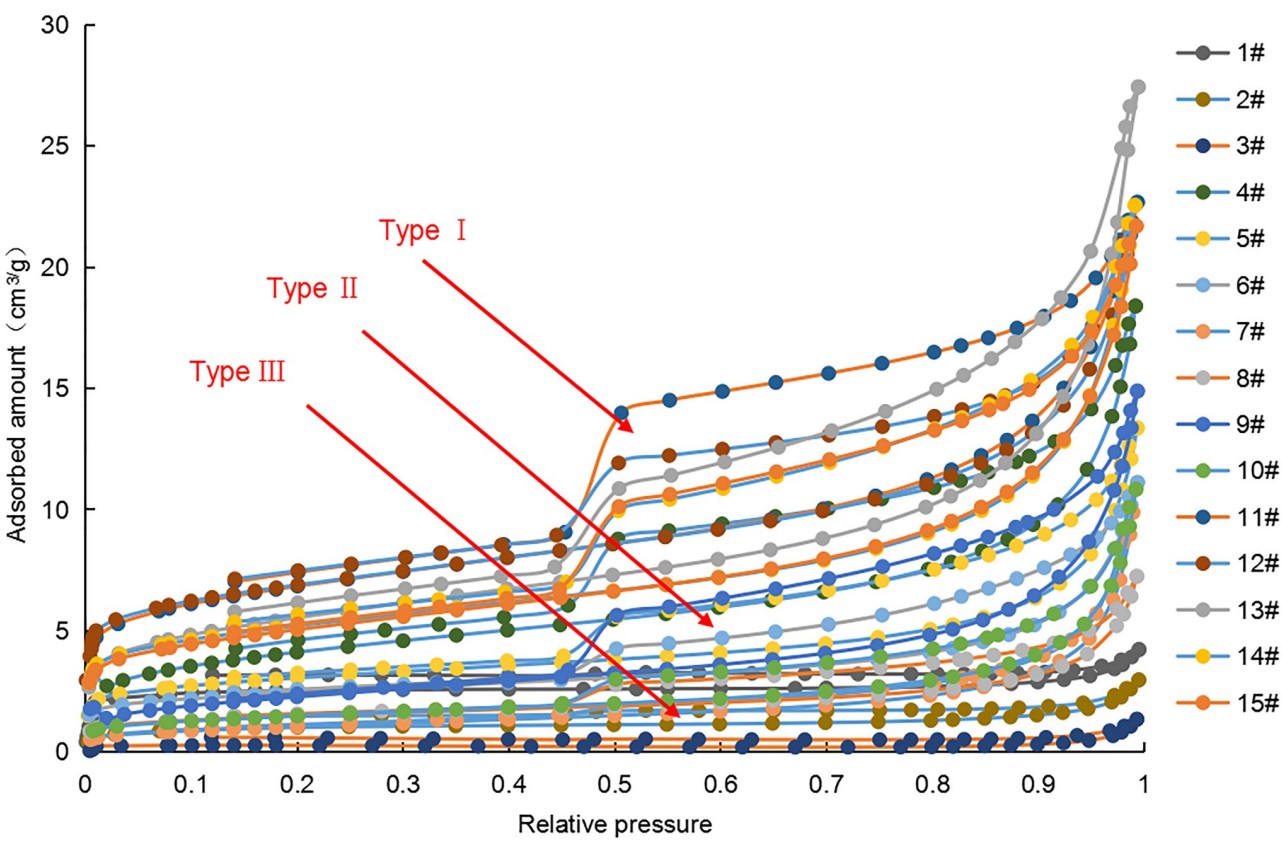

**Fig 8. Isotherm adsorption and desorption curves for different samples using low-temperature nitrogen.**

mainly micropores. In the thick organic-rich shale samples of the Middle and Lower Longtan members, organic matter is distributed only sporadically among detrital particles, and isolated high round organic pores are developed (Fig 6G), with relatively small pore size. Inorganic pores are mainly interlayer pores of clay minerals, and a few mineral-grain margin fractures occasionally developed. The pore size is mainly micropores and mesopores, presenting a bimodal distribution with peak values at about 10 nm and 50 nm respectively (Fig 9C). Some micron storage spaces are occasionally developed. Micropores are the main contributor to specific surface area, whereas mesopores and macropores are the main contributors to pore volume (Fig 9D). In the shale samples with organic matter from the interbedded sandstone-mudstone of the Middle and Lower Longtan members, inorganic pores and microfractures are relatively well developed, but organic pores are relatively scarce. Organic matter is distributed in clumps between clastic particles or clay minerals, and some organic pores are connected (Fig 6H and 6I). The specific surface area of the organic pores can provide specific surface for hydrocarbon adsorption. Inorganic pores include intercrystal pores of clay minerals, interparticle pores of quartz and feldspar, and dissolution pores of carbonate minerals. Silt particles exhibit more line contact. Clay minerals mostly occur between silt particles, forming numerous interlayer pores of clay minerals with large pore volumes and providing good storage space for free hydrocarbons. Microfractures are mainly apparent as mineral shrinkage and dissolution pore fractures caused by diagenesis (Fig 6J and 6L). The microfractures provide space for shale gas storage and can also serve as channels for shale gas migration. The pore size distribution generally shows multi-peak characteristics, with peaks at 5 nm, 50–150 nm, and

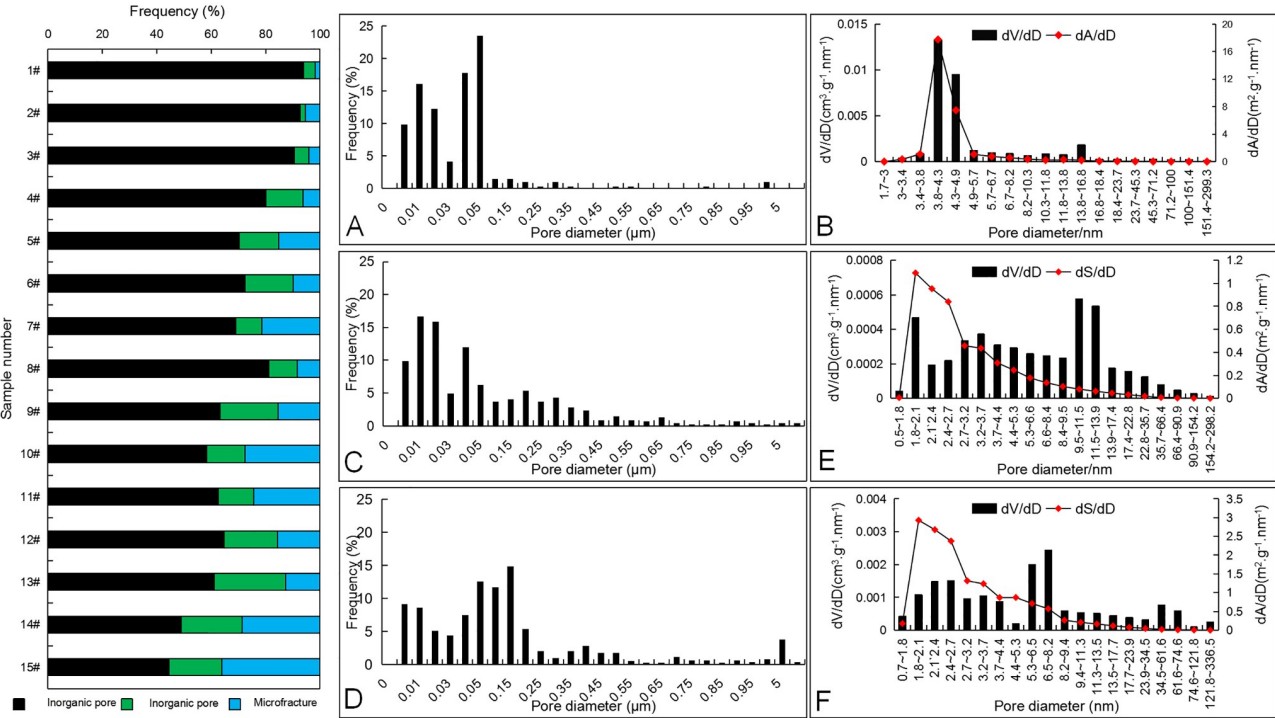

**Fig 9. Pore types and structure characteristic parameters in different shales by SEM and Image-pro Plus.** Aperture distribution histogram of all pores in (A) Sample 1# from Upper Longtan member; (B) Sample 6# from Middle Longtan member; (C) Sample 13# from Lower Longtan member. Distribution histogram of specific pore volume and specific surface area for (D) Sample 1# from Upper Longtan member; (E) Sample 6# from Middle Longtan member; (F) Sample 13# from Lower Longtan member.

1–5 μm respectively (Fig 9E). Fig 9F shows that the specific surface area of organic-rich shale samples in the Middle and Lower Longtan members is mostly contributed by micropores, but that mesopores and mesopores make the largest contribution to pore volume.

From bottom to top, organic pores in the organic-rich shale of the Longtan Formation are poorly developed as a whole, and the storage space consists mainly of inorganic pores and microfractures. With the evolution of shale lithofacies from silty shale to clay shale and then to shale with silty or calcareous shale, the proportions of interparticle pores of clay minerals and of dissolution pores decrease gradually, and the pore size decreases as well. In the silty shale of the Lower Longtan member, inorganic pores and microfractures are relatively well developed, with large pore size (Fig 9).

The development of reservoir storage space is controlled by different mineral compositions in shale reservoirs. Quartz and clay minerals, as the main components of shale, have different degrees of influence on the development of storage space [10]. Statistics revealed that the quartz contents of different shale samples from the Longtan Formation in the study area have low correlation with specific surface area (Fig 10A), and positive correlation with specific pore volume (Fig 10B). Clay minerals are positively correlated with the specific surface area of different samples (Fig 10C), but not obviously with specific pore volume (Fig 10D). The reason for this is that brittle minerals such as quartz in samples from marine-continental transitional sedimentary environments often show good compaction resistance, which makes the rocks preserve some primary pores under compaction (Fig 10E and 10F) [43]. The brittleness index (BI) of different samples was calculated based on mineral composition: BI = (quartz / (quartz + clay minerals + calcareous minerals))×100. The results show that the proportion of

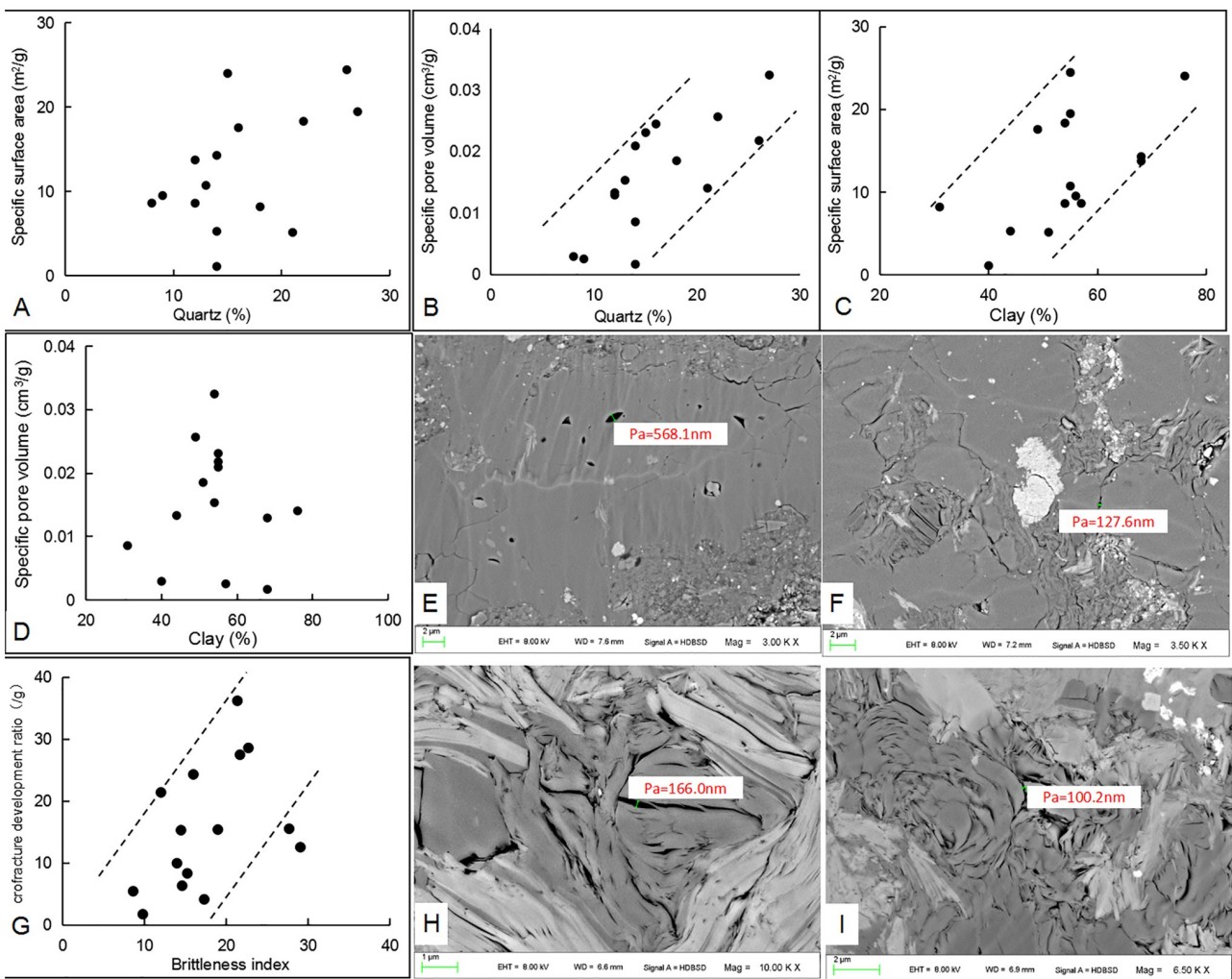

**Fig 10. Relationships between different mineral components (quartz, clay minerals) and pore volume, specific surface area, and microfracture development ratio.** (A) Relationship between quartz and specific surface area; (B) Relationship between quartz and specific pore volume; (C) Relationship between clay minerals and specific surface area; (D) Relationship between clay minerals and specific pore volume; (E) An irregular pore and microcrack in microgranular quartz (×3000); (F) Particle margin fractures in microcrystalline quartz (siliceous) (×3500); (G) Correlation between microfracture development ratio and brittleness index; (H) Interbedded fracture in lamellar andreattite aggregation (×10000); (I) Interlayer pores in alternately distributed lamellar andreattite and chlorite aggregation (×6500).

microfractures in all storage spaces increases with increasing BI in different samples (Fig 10G), and that the presence of quartz plays a positive role in the development of microfractures. Clay minerals are mainly chlorite and andreattite. Lamellar silicate minerals are often found in lamellar andreattite and chlorite aggregate, with interlayer pores and large surface due to fine particles (Fig 10H and 10I).

In the Middle and Lower Longtan members, the lithological assemblage is sandstone, mudstone, and limestone interbedded with coal seams. A coal seam, as an organic aggregation, has much higher gas-generation capacity than shale, and coal gas will migrate to adjacent siltstone and shale to accumulate and increase gas-bearing capacity [10]. In addition, a high content of brittle minerals is conducive to generation of natural or induced fractures, which in turn are conducive to desorption, seepage, accumulation, and exploitation of

natural gas adsorbed in shale. One of the main forms of shale gas is adsorbed by clay minerals. As a good carrier of shale gas migration and accumulation, clay minerals also play an important role in shale gas occurrence, migration, and accumulation. Part of the organic matter is distributed microscopically among the clay mineral grains and adsorbs gas molecules during shale gas generation. Illite and andreattite in clay minerals can promote kerogen cracking and play a catalytic role in organic gas generation. In addition to a high content of brittle minerals, organic-rich shale in the lithological assemblage of high-frequency interbedded sand-mud-coal has developed fissures and laminations. The large lithological differences between layers causes the stress on the weak surface to increase, and many horizontal fractures can easily form under the action of external forces, which promotes the radial extension of hydraulic fractures.

## 5. Conclusions

1. The Longtan Formation in the study area belongs to a marine-continental transitional facies with complex lithology that consists mainly of developed delta and tidal flat-lagoon sedimentary systems. Combined with the characteristics of different lithological assemblages in the marine-continental transitional sedimentary environment, the Longtan Formation can be further divided into the Lower, Middle, and Upper Longtan members. The Upper Longtan member is dominated by deltaic sedimentation, with cross- and parallel bedding, and contains plant fossils. The sedimentary environment of the Middle and Lower Longtan members was dominated by tidal flats and lagoons with some swamps, and featured wavy, horizontal, and slight cross-bedding, and contains banded and nodular siderite developed together. The main lithology includes thick shale interbedded with limestone, siltstone, and interbedded shale, siltstone and limestone, with thick coal seams.

2. The kerogen types of different lithological assemblages in shale are type $II_2$-III, with a mean vitrinite reflectance of 2.66%, indicating favorable thermal maturity for shale gas exploration. Inorganic pores are the main storage space in a shale reservoir, with a few organic pores and microfractures. The difference between the hysteresis loops in low-pressure $N_2$ adsorption and desorption test results indicates three types of storage space.

3. The storage characteristics of shale reservoirs in different lithological assemblages have been compared and analyzed. The organic pores of organic-rich shale in the Longtan Formation are not well developed as a whole, and the types of storage space are mainly inorganic pores and micro-fractures. With the evolution of shale lithofacies from silty shale to clay shale and then to shale with silty or calcareous shale, the proportions of interparticle pores of clay minerals and of dissolution pores decrease gradually, and the pore size decreases as well. In the silty shale of the Lower Longtan member, inorganic pores and microfractures are relatively developed, with large pore size. Clay minerals play a positive role in the specific surface area of a sample, whereas quartz has a positive influence on the protection of specific pore volume.

## Supporting information

**S1 Data.**
(XLSX)

**S2 Data.**
(XLSX)

**S3 Data.**
(XLSX)

**S4 Data.**
(XLSX)

**S5 Data.**
(XLSX)

**S6 Data.**
(XLSX)

## Acknowledgments

The authors greatly acknowledge the anonymous reviewers for their insightful and thorough reviews for their great help.

## Author Contributions

**Conceptualization:** Qian Cao.

**Data curation:** Qian Cao.

**Formal analysis:** Yan Liu, Ke Jiang.

**Methodology:** Xin Ye, Pengwei Wang.

**Project administration:** Pengwei Wang.

**Resources:** Yan Liu.

**Writing – original draft:** Qian Cao.

**Writing – review & editing:** Qian Cao.

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
