## [Decision Letter · Decision Letter 0]

9 Feb 2022

PONE-D-21-40287Effect of different lithological assemblages on oil shale reservoir properties in the Permian Longtan Formation, southeastern SichuanPLOS ONE

Dear Dr. CAO,

Thank you for submitting your manuscript to PLOS ONE. After careful consideration, we feel that it has merit but does not fully meet PLOS ONE’s publication criteria as it currently stands. Therefore, we invite you to submit a revised version of the manuscript that addresses the points raised during the review process.

We look forward to receiving your revised manuscript.

Kind regards,

Paola Cipollari

Academic Editor

PLOS ONE

Journal Requirements:

2. n your Methods section, please provide additional information regarding the permits you obtained to collect samples for the present study. Please ensure you have included the full name of the authority that approved the field site access and, if no permits were required, a brief statement explaining why.

5. Please amend either the title on the online submission form (via Edit Submission) or the title in the manuscript so that they are identical.

6. Please amend the manuscript submission data (via Edit Submission) to include author Pengwei Wang.

7. Please remove your figures from within your manuscript file, leaving only the individual TIFF/EPS image files, uploaded separately.  These will be automatically included in the reviewers’ PDF.

8 Please include your tables as part of your main manuscript and remove the individual files. Please note that supplementary tables (should remain/ be uploaded) as separate "supporting information" files.

Additional Editor Comments:

This is an interesting integrative paper combining several different methodologies to define the relationships between the depositional environment, and the related lithologies, and the development of gas storage space. I appreciate the valuable contribution of the data presented here. However, certain issues need to be worked on before that paper can be published. The main issue is the writing that needs to be carefully edited throughout. I would suggest consulting a native speaker for a proper revision of English.

More problematic is the rather poorly developed introduction; it needs more attention for readers unfamiliar with the geology of this region to properly understand. The Reviewer highlights several significant issues throughout that need attention.

I noted the incompleteness of the figure captions, as did the reviewer, and a lack of accuracy in the drafting of the text (e.g., uncited figures).

According with the reviewer, I suggest operating major revisions, following the reviewer suggestions, to make the paper suitable for the publication.

Reviewers' comments:

Reviewer's Responses to Questions

**Comments to the Author**

1. Is the manuscript technically sound, and do the data support the conclusions?

Reviewer #1: Yes

2. Has the statistical analysis been performed appropriately and rigorously? 

Reviewer #1: Yes

3. Have the authors made all data underlying the findings in their manuscript fully available?

Reviewer #1: Yes

4. Is the manuscript presented in an intelligible fashion and written in standard English?

Reviewer #1: No

5. Review Comments to the Author

Reviewer #1: Dear Editor,

Thank you for the opportunity to review the manuscript “Effect of different lithological assemblages on oil shale reservoir properties in the Permian Longtan Formation, southeastern Sichuan” (Manuscript: PONE-D-21-40287) by Qian Cao and Co-authors. This research focuses on the shale reservoir characteristic from different lithological assemblages of the Permian Longtan Formation in southeastern Sichuan. The authors seek to improve our understanding of the oil shale reservoir properties in the Longtan Formation by providing X-ray diffractometry (XRD), organic geochemistry, organic maceral identification, Ar-ion polishing, scanning electron microscopy (SEM), and N2 adsorption testing of samples collected for the abovementioned Formation. Overall, the quality of this research is good, and it will likely be of interest to a board audience.

In my opinion, this article in its current conditions required major revisions before to be accepted for publication on the PLOS ONE. I have several suggestions to improve the readability, accessibility, and impact of this research, and many of the more specific points are in the main reviewed space.

• I realize that the authors have a misleading in the meaning of Formation, i.e., in the Abstract section, the authors state that “the lithological development divides the Longtan Formation into lower (swamp), middle (tidal flat/lagoon) and upper (delta) sub-members” whereas, on the manuscript (section 3.1), the authors state that “the differences in the lithological assemblages in showed that Longtan Formation could be further divided into the Lower, Middle and Upper Longtan Formations.” From a stratigraphy point of view, a formation can be divided into members and grouped together in groups. Please, modify to lower, middle, and upper members.

• The main issue is the writing style that is quite poor in places and needs careful and thorough review to improve the clarity, conciseness, grammar, sentence structure, and organization. I accept that the authors may not be native English speakers, but the text would benefit from additional editing. I suggest the authors significantly polish the English or get the help of a native speaker. Some sentences are too long and entire paragraphs are too confusing for the reader. It should be made both more concise and sharper.

• The introduction should provide a better geographical context for readers not familiar with the study area.

• I encourage the authors to consider citing the figures of the manuscript (i.e., figure 7 is not cited at all on the manuscript).

• Figure captions are inadequate. Figure captions should be comprehensive but concise; they should clearly describe the contents of the figure; they should draw attention to key features in the figure. Figure captions should provide enough information so that the reader can easily review the figure without referring to the text. So, I would ask the authors to rewrite the figure captions.

• In figure 1, the authors should add a general map to better show the southeaster Block of the Sichuan Basin in Southern China and cite it in the introduction (section 1). In figure 1b, I motivate the authors to add the samples label of the 15 collected samples to show the exact stratigraphic position of each sample in the well-log. Always in figure 1b, it is too difficult to see the lithologies in the well-log.

6. PLOS authors have the option to publish the peer review history of their article (what does this mean?). If published, this will include your full peer review and any attached files.

Reviewer #1: **Yes: **Anas Abbassi

---

## [Author Response · Author response to Decision Letter 0]

5 May 2022

Dear Paola Cipollari and Reviewers,

On behalf of my co-authors, we thank you very much for giving us an opportunity to revise our manuscript, we appreciate editor and reviewers very much for their positive and constructive comments and suggestions on our manuscript entitled “properties in the Permian Longtan Formation, southeastern Sichuan Basin: Case study of Well X1”. (ID: PONE-D-21-40287). 

We have studied reviewers’ comments carefully and have made revisions that are marked in red in the paper. We have tried our best to revise our manuscript according to the comments. Attached please find the revised version, which we would like to submit for your kind consideration.

We would like to express our great appreciation to you and the reviewers’ comments on our paper. Looking forward to hearing from you.

Thank you and best regards.

Yours sincerely,

Dr. Qian Cao

State Key Laboratory of Shale Oil and Gas Enrichment Mechanisms and Effective Development

Response to Reviewers

1. I realize that the authors have a misleading in the meaning of Formation, i.e., in the Abstract section, the authors state that “the lithological development divides the Longtan Formation into lower (swamp), middle (tidal flat/lagoon) and upper (delta) sub-members” whereas, on the manuscript (section 3.1), the authors state that “the differences in the lithological assemblages in showed that Longtan Formation could be further divided into the Lower, Middle and Upper Longtan Formations.” From a stratigraphy point of view, a formation can be divided into members and grouped together in groups. Please, modify to lower, middle, and upper members.

Reply: By relearning the meaning of “Formation” that a formation can be divided into members and grouped together in groups, the description of Longtan Formation has been reorganized from a stratigraphy point of view in the paper. The Longtan Formation was divided into the lower, middle, and upper members due to the different lithological development. This part of the description has also been modified throughout the article. And all the amendments are highlighted in red in the revised manuscript.

2. The main issue is the writing style that is quite poor in places and needs careful and thorough review to improve the clarity, conciseness, grammar, sentence structure, and organization. I accept that the authors may not be native English speakers, but the text would benefit from additional editing. I suggest the authors significantly polish the English or get the help of a native speaker. Some sentences are too long and entire paragraphs are too confusing for the reader. It should be made both more concise and sharper.

Reply: The language of our manuscript have been refined and polished by International Science Editing. The supporting documents are as follows:

3. The introduction should provide a better geographical context for readers not familiar with the study area.

Reply: We have described the geographical context of the study area in detail in Chapter 2.1 of the article, including the changes of sedimentary environment, stratigraphic distribution, etc. line 40-46, line 51-64, and line 82-93.

4. I encourage the authors to consider citing the figures of the manuscript (i.e., figure 7 is not cited at all on the manuscript).

Reply: We have checked the descriptions of all graphs and tables in the article in detail, and each graph and table is reasonably referenced.

5. Figure captions are inadequate. Figure captions should be comprehensive but concise; they should clearly describe the contents of the figure; they should draw attention to key features in the figure. Figure captions should provide enough information so that the reader can easily review the figure without referring to the text. So, I would ask the authors to rewrite the figure captions.

Reply: We have tried our best to rewrite the figure captions according to the comments, including Figures 1, 2, 5, 6, 10 in Supporting information, and line 512 - 583.

6. In figure 1, the authors should add a general map to better show the southeaster Block of the Sichuan Basin in Southern China and cite it in the introduction (section 1). In figure 1b, I motivate the authors to add the samples label of the 15 collected samples to show the exact stratigraphic position of each sample in the well-log. Always in figure 1b, it is too difficult to see the lithologies in the well-log.

Reply: We have added a general map to identify the location of Sichuan Basin, this figure is explained in the introduction (line 40-46, and line 53-55). Besides, the lithology and other information of the sample are also recorded when collecting these samples, and the lithologies labels of these 15 samples have been identified in Figure 1B. It is hoped that this description will enable readers to clearly see the information of the collected samples.

We tried our best to improve the manuscript and made some changes in the manuscript. These changes will not influence the content and framework of the paper. And here we have marked the changes in red in revised paper.

We appreciate for editors and reviewers’ warm work earnestly, and hope that the correction will meet with approval.

Once again, thank you very much for your comments and suggestions.

---

## [Editor Report · Decision Letter 1]

22 Jun 2022

Effect of different lithological assemblages on shale reservoir properties in the Permian Longtan Formation, southeastern Sichuan Basin: Case study of Well X1

PONE-D-21-40287R1

Dear Dr. Qian CAO,

We’re pleased to inform you that your manuscript has been judged scientifically suitable for publication and will be formally accepted for publication once it meets all outstanding technical requirements.

Kind regards,

Paola Cipollari

Academic Editor

PLOS ONE
---

## [Editor Report · Acceptance letter]

4 Aug 2022

PONE-D-21-40287R1 

Effect of different lithological assemblages on shale reservoir properties in the Permian Longtan Formation, southeastern Sichuan Basin: Case study of Well X1 

Dear Dr. CAO:

I'm pleased to inform you that your manuscript has been deemed suitable for publication in PLOS ONE. Congratulations! Your manuscript is now with our production department. 

Kind regards, 

on behalf of

Dr. Paola Cipollari 

Academic Editor

PLOS ONE